# Murburn Bioenergetics and “Origins–Sustenance–Termination–Evolution of Life”: Emergence of Intelligence from a Network of Molecules, Unbound Ions, Radicals and Radiations

**DOI:** 10.3390/ijms26157542

**Published:** 2025-08-05

**Authors:** Laurent Jaeken, Kelath Murali Manoj

**Affiliations:** 1Department of Industrial Sciences and Technology, Karel de Grote-Hogeschool, Antwerp University Association, Salesianenlaan 90, 2660 Antwerpen, Belgium; 2Satyamjayatu: The Science & Ethics Foundation, Shoranur-2, Palakkad 679122, Kerala, India; 3Amrita School of Artificial Intelligence, Coimbatore, Amrita Vishwa Vidyapeetham, Amritanagar, Ettimadai 641112, Tamil Nadu, India

**Keywords:** mitochondria, murburn concept, coacervation, diffusible reactive (oxygen) species (DRS/DROS/ROS), coherence, origin of life, aging, cancer, bioenergetics

## Abstract

The paradigm-shift idea of murburn concept is no hypothesis but developed directly from fundamental facts of cellular/ecological existence. Murburn involves spontaneous and stochastic interactions (mediated by murzymes) amongst the molecules and unbound ions of cells. It leads to **e**ffective **c**harge **s**eparation (ECS) and formation/recruitment of **d**iffusible **r**eactive **s**pecies (DRS, like radicals whose reactions enable ATP-synthesis and thermogenesis) and emission of radiations (UV/Vis to ELF). These processes also lead to a **c**hemo-**e**lectromagnetic **m**atrix (CEM), ascertaining that living cell/organism react/function as a coherent unit. Murburn concept propounds the true utility of oxygen: generating DRS (with catalytic and electrical properties) on the way to becoming water, the life solvent, and ultimately also leading to phase-based macroscopic homeostatic outcomes. Such a layout enables cells to become **s**imple **c**hemical **e**ngines (SCEs) with **p**owering, **c**oherence, **h**omeostasis, **e**lectro-**m**echanical and **s**ensing–response (PCHEMS; life’s short-term “intelligence”) abilities. In the current review, we discuss the coacervate nature of cells and dwell upon the ways and contexts in which various radiations (either incident or endogenously generated) could interact in the new scheme of cellular function. Presenting comparative evidence/arguments and listing of systems with murburn models, we argue that the new perceptions explain life processes better and urge the community to urgently adopt murburn bioenergetics and adapt to its views. Further, we touch upon some distinct scientific and sociological contexts with respect to the outreach of murburn concept. It is envisaged that greater awareness of murburn could enhance the longevity and quality of life and afford better approaches to therapies.

## 1. Introduction

Do we have a fair grasp of the principles behind **o**rigins, **s**ustenance, **t**ermination and **e**volution **o**f **l**ife (OSTEoL) and how aging or degenerative diseases are based on them? Although we have come a significant way to understand life processes, there are many lacunae and misperceptions, particularly in bioenergetics, owing to which we cannot address causes, prophylaxis and therapy in a clear/holistic manner. This review presents a novel “murburn” bioenergetics framework that resolves prevailing confusions. 

Research leading to murburn (“***mur***ed ***burn***ing” or closed oxidative process) concept [1] disclosed the previously ignored, yet unavoidable, central physiological function of diffusible reactive (oxygen) species or DRS/DROS (which could be nitrogen or sulfur or hydrogen centered too in anaerobic systems; so DRS is a globally applicable acronym, with suitable derivations thereof!) at low concentrations, namely that of coupling factors in numerous redox processes, like oxidative-/photo-phosphorylation, leading to direct *in fluid* ATP-synthesis by *exergonic* radical reactions. (We emphasize the term “diffusible” and do not use ROS alone, to differentiate it from enzyme-bound ROS or BROS, such as Compound I of hemeproteins.) The formation of oxygen-centric DRS (anions/radicals) near proteins/membrane surfaces also leads to effective charge separation (ECS), serving as the origin of “*cellular electrical activity*”. The stochastic interactive chemical equilibriums of DRS with components in milieu (ADP, Pi and others) are characterized by an observable new type of stochastic, hormetic and idiosyncratic “*outside the active site*” mechanism/kinetics. In conjunction with coacervate organization, murburn could maintain discrete ion and solute differentials across “islets” of different phases without ion-pumping or deterministic electron transport chains (ETCs). Murburn-type reactions within pre-biotic coacervates afforded powering, coherence, homeostasis, electro-mechanical features and sensing/responding abilities (PCHEMS: the “instant” functions of life [1]), later supplemented and propagated by genetic means in cellular systems. As murburn (with DRS-ECS as core physiology) and coacervation rooted cellular PCHEMS since life’s origin and continued to sustain it thereafter, normal aging, disease, evolution and termination must be understood within this framework (although these functions became later associated with complex membranes and central dogma’s “deterministic” exertion). Also, radiative emissions associated with DRS reactions and ECS currents are the primary actors in the build-up of the chemo-electromagnetic matrix (CEM, “cement-like binder of life”) that makes a living structure and its activities act as a single unit.

This write-up highlights diverse aspects of the murburn concept above [1], a fundamental revolution unraveled over the last decade, and discusses novel approaches for addressing various (patho)physiological scenarios, giving a new perspective on holistic organellar (e.g., mitochondrial) or cellular function.

## 2. Conventional Views of Life-Sustaining Mechanisms and Associated Issues

The organelle/cellular membranes are seen as “sentient sentinels”, selectively pumping and transporting diverse molecules/ions in and out via high-throughput stoichiometric mechanisms [1]. Although the classical views recognize coacervation to be an important step in life’s origin, it is surprisingly neglected in theoretical models explaining various physiological processes. Since the early 20th century, the laws of dilute solutions, active-site (affinity-based selectivity/specificity) enzyme catalysis and protein-conformational changes have been used to explain key aspects of cellular functioning [1]. Central dogma describes the core information flow with genetic expression regulating proteins and their concentrations, which in turn deterministically control metabolic rates and cellular composition. Most importantly, cellular powering (which supports other crucial functions!) is thought to rely on the generation/maintenance of ion-gradients across the so-called “bioenergetic membranes”. Under the acclaimed schema, ATP is seen as the global bioenergetic currency whose hydrolysis enables cells to carry out otherwise unfavorable and vitally deterministic processes. Mitochondria and chloroplasts in eukaryotes (where ATP synthesis occurs) are accepted to have originated from prokaryote precursors by means of endosymbiosis [1].

Several aspects of this model remain puzzling. For instance: How could the primordial “pre-life” entity have the ability to control its inside as well as outside for maintaining trans-membrane potential (TMP; including proton-motive force or *pmf*) before such a complex membrane could have ever developed? How can contemporary cells continue to do so, even with a membrane? Why do we acutely need oxygen for life activities, if it only serves as a “terminal electron acceptor” (TEA)? If ROS are toxic, wasteful products of metabolism, why did the redox proteins not acquire structural features to minimize their production and how can a radical like nitric oxide serve as a systemic messenger? How could cells function as a single, whole, self-regulating unit at instant timescales (wherein genetic control cannot be exercised)? We attempt to answer such queries on OSTEoL (the basic skeletal scaffold of “perpetual life intelligence”) with the umbrella of ideas espoused under murburn concept.

## 3. Life’s Origins Must Have Used DRS

Some salient aspects of life’s origins are discussed here for addressing its causality and continuum in time, from “pre-membrane-bound soups of yore” to “membrane-bound life systems of today”. In the anaerobic environment of the young Earth, CH_4_, NH_3_, HCN, H_2_O and H_2_S were probably the prominent molecular species to start the organic soup. Without DRS, there would have been no hydroxy, keto, carboxylic and amino acids, even under anaerobic conditions. The oxygen atoms in these acids were most probably incorporated via the generation of DRS from the solvent of life, water. This process was likely catalyzed by Fe^3+^/Fe^2+^ cycles (as testified by banded iron formations containing both ionic forms) and powered by UV-Vis light (the first photosynthesis). Unlike now, within a reducing environment, Fe^2+^ would have been the more stable form. During the day, photo-oxidation to Fe^3+^ would make conditions oxidative, whereas, during the night, they would be reducing [1]. Fe^3+^ can photoreact with water generating DROS and O_2_ [2], which continues because the downstream reactions are exothermic (resulting in thermodynamic pull; a feature of murburn!). This circadian iron cycle with alternating oxidative and reductive abilities might be the earliest precursor of catabolic–anabolic cycling, a fundamental hallmark of life [1]. These fundamental reactions (Box 1) via DRS enabled both of the components of the solvent (hydrogen and oxygen atoms) to participate in *equilibriums* (life’s processes!), forming the first type of auto-regulative logic with respect to chemico–physical transformations. The benefit of having oxygen (via DROS) as the oxidizer was the facilitated delivery of gaseous reactant and voiding of the gaseous product, CO_2_. Further, it enabled the full redox window of ~1.2 V (approximately −400 to +800 mV) for conducting cellular redox processes, enabling the synthesis and functionalizing of diverse biomolecules and transition to an aerobic environment.

Box 1Simple bimolecular DRS-involving cascades at the origins of life.
*Initially, DRS production is from water:*
Fe^3+^ + H_2_O + hv →  Fe^2+^ + H^+^ + HO* (188 kJ/mol ≈ 637 nm photon)HO* + HO* →  H_2_O_2_ (−185.8 kJ/mol)
*Fe^2+^/Fe^3+^ cycling of peroxide via Fenton-type reactions lead to DRS and O_2_:*
Fe^2+^ + H_2_O_2_ →  Fe^3+^ + OH^−^ + HO*Fe^3+^ + H_2_O_2_ →  Fe^2+^ + H^+^ + HOO*-------------------------------------------------------2H_2_O_2_ →  H_2_O + HO* + HOO* (60.1 kJ/mol)Fe^3+^ + HOO* →  Fe^2+^ + H^+^ + O_2_ (−61.4 kJ/mol)
**---------------------------------------------------------------**
Fe^3+^ + 2H_2_O_2_ →  Fe^2+^ + H_2_O + HO* + H^+^ + O_2_ (−1.3 kJ/mol)
*Later, the DRS generated can undergo Haber–Weiss-type reactions forming O_2_:*
H_2_O_2_ + HOO* →  HO* + H_2_O + O_2_ (−64.5 kJ/mol)HOO* + HOO* →  H_2_O_2_ + O_2_ (−124.7 kJ/mol)HOO* + HO* →  H_2_O + O_2_ (−250.1 kJ/mol)

Thioesters were spontaneously formed in the initial phases, owing to the facile nature of the reaction from H_2_S; in which case, the intermediary is DRSS, sulfur-centered DRS [3]. In present-day metabolism thioesters (like acetyl-CoA) are still important intermediates, remnants of the early origin. DRSS could have played essential roles in early oligomerization processes [3]. In early nitrogen metabolism DRNS might have been involved (the molecular messenger of nitric oxide, NO, still is). At later times, when nucleobases, ribose and phosphate became available, ADP/ATP could have formed in situ from AMP via simple bimolecular reactions by sequential phosphorylations, like: AMP/ADP + Pi + DRS → ADP/ATP + H_2_O/H_2_O_2_ + heat [1]. They can be executed in vitro in a dense phase without enzymes or membranes, and in vivo indications ensure that this simple, versatile, highly advantageous mechanism has persisted [1,4]. The reactions given above are the primordial murburn-type processes leading to life and sustained to date!

## 4. Murburn Concept: A Constitutive and Integrative Pillar of Life, from Its Very Origins

From the “free-burning” activities in the early times, the reactions became milder and “mured”. Murburn activity generates/utilizes ECS and non-deterministic (probabilistic/stochastic) reaction chemistry through formation/utilization of DRS [1,4]. It is a tangible and viable purview for explaining real-time PCHEMS of cells, replacing the classical ETC and Mitchell’s proton-pump proposal.

The Mitchellian bioenergetic explanation requires the establishment of *pmf* using serial arrays of ETCs that are presumed to serve as proton pumps. Not just that, the classical model also requires Complex V to carry out “chemiosmotic rotational ATP-synthesis” (CRAS) through the sophisticated orchestration of both *pmf* generation and its harvest by “*intelligent system coordinating multiple components spanning three phases*” (Figure 1)! (This requires concerted working of dispersed proteins, leading to temporally staggered outcomes of building and dissipation of *pmf*. However, such coordinated outcomes and voltage fluctuations are not empirically observed!) Clearly, such explanations present “*irreducibly-complex, speculative and gambit-logic*” [1] that preclude step-wise evolutionary developments and have little basis in thermodynamics or reality. Quite simply, there was/is little scope for a cell to control its internal and external environments (both!) and manipulate it temporally/perpetually for generating/harvesting TMP/*pmf* by “electrogenic cation pumping” or avail some cyclic ion exchange logic [1,4]; all allegedly by a “highly intelligent” membrane!

### 4.1. Fighting Against Stark Reality?

When it is easily discernible that the cellular milieu is a complex colloid (evident by breaking open a bird’s egg!) and should be described by the laws of colloid physico-chemistry, cell physiologists have traditionally used Nernst’s laws for calculating ion gradients using the laws of “ideal” dilute solutions. This is when modern microscopy and spectroscopy provide ample support for the colloidal/complex nature of the cellular milieu [5,6,7].

The root cause of the longstanding mischaracterization of taking the cellular milieu as a solution goes back to the unrealistic extrapolations from a limited experiment by the influential Nobel Laureate Archibald Hill in 1930 [8]. He had found that the cell versus environment partition coefficient for urea was close to 1, implying that the cellular interior was just like the exterior! However, other substances such as D-mannitol (0.227) or sucrose (0.132) and over 25 others tested gave lower values, corroborating the coacervate nature of cells [5,6,7,9]. Schrödinger [10] had falsified Hill’s arbitration in an insurmountable way, stating that it would imply that the finest control life could have over its working then has a lower limit dictated by *k*T. Further, while cyanobacteria are aprotic, the mitochondrial matrix contains finger-countable protons [1,4,5] (Box 2)! How could such a system ever serve tens of thousands of redox complexes (proton pumps) in the cells/organelles? How could so few protons ever build a *pmf* able to energize ATP synthesis, when water hydrolysis requires >79 kJ/mol (for generating protons)? And how could Complex V synthesize ATP with a non-existent *pmf* via an esterification mechanism (fusing two negatively charged ionic species), overcoming a ΔG of +36 kJ/mol (K_eq_ ~5 × 10^−7^), at high physiological ATP/ADP ratios on Walker sites (which have higher affinity for ATP and favor ATP hydrolysis over ATP synthesis by a factor of 10^8^) [4,11]?

Box 2Finger-countable protons in prokaryotes/bioenergetic organelles!***Small cyanobacterium*** →  10^−16^ L (non-adjusted for internal components) volume × 10^−8^ moles/L concentration of protons × 6.023 × 10^23^ protons/mole = **≤1*****Average mitochondrion*** →  2 × 10^−16^ L (non-adjusted for internal components) volume × 5 × 10^−8^ moles/L concentration of protons × 6.023 × 10^23^ protons/mole = **6*****Average bacterium*** →  4 × 10^−16^ L (non-adjusted for internal components) volume × 4 × 10^−8^ moles/L concentration of protons × 6.023 × 10^23^ protons/mole = **10*****Average stroma of chloroplast*** →  3 × 10^−17^ L (adjusted volume, https://doi.org/10.1016/j.febslet.2013.05.046) × 5 × 10^−7^ moles/L concentration of protons × 6.023 × 10^23^ protons/mole = **9**

### 4.2. Wishful Thinking?

Another error was an aesthetic/deterministic *assumption* by David Keilin [12], who had pioneered the characterization of cytochromes. The reasoning proposed that electron transport must be organized as a *serial* system (a *chain*) with oxygen serving as a terminal electron acceptor at cytochrome oxidase to *prevent* DROS formation. But the fact that DROS are formed at each of the redox complexes is testament to the reason that one-electron reduction of oxygen is very spontaneous (Δ*G* = −250 kJ/mol) and oxygen is highly mobile (capable of facile diffusion at the approximate value of ~10^2^ Å/μs). Thus, intra/intermolecular deterministic e-transfers across several Angstroms would be unfeasible in micro- to milli- second timescales (as necessary for ETCs)! Simply put, the serial ETC lacks kinetic and probabilistic feasibility. 

### 4.3. The Turnaround!

By 2018–2019, the untenable and redundant aspects of the classical “ETC–proton pump–*pmf*–CRAS” paradigm were conclusively pointed out via multiple analytical angles, with more than 30 incisive comparative arguments [4,13,14]. Further, after the in vitro *in-fluid* demonstration of ATP synthesis from its precursors using DRS [4], with dozens of predictive/comparative queries on available evidence [4], the murburn model was established as the well-substantiated model for oxidative phosphorylation [11,13,14,15,16]. Murburn energy metabolism allows for facile transitions from earlier anaerobic/prokaryotic systems to later aerobic/eukaryotic systems [1]. It obviates the cell/organelle to possess the ability to detect/control components inside/outside (for homeostasis and TMP/*pmf*) (Figure 2). Modern experimental research has conclusively shown that DRS are effectively formed at each of the respiratory redox complexes, their production rate is correlated with respiratory rate, an adequate O_2_ supply keeps DRS concentrations low, and DRS have beneficial effects at low concentrations [1,4,17]. In-fluid or interfacial synthesis of ATP (according to murburn bioenergetics) is facile, via exergonic bimolecular reactions. It makes redox complexes the essential ATP-synthases, for which there is detailed structural evidence. These proteins possess intra-protein small cavities, where redox centers can donate single electrons to O_2_ initially producing O_2_*^−^ (and secondarily other diffusible reactive species). In conjunction with ADP-presenting sites pointed out on the various respiratory membrane protein complexes, these realities enable effective activation of phosphate groups on ADP/Pi [1,4,10,11]. All reactions going on there are exergonic (Box 3).

Box 3Kinetically viable murburn reactions.
*Non-radical reactions*
NADH + ^1^O_2_ (+H^+^) →  NAD^+^ + H_2_O_2_ (−676 kJ/mol)QH_2_ + ^1^O_2_ →  Q + H_2_O_2_ (−154 kJ/mol)
*Radical-generating reactions*
O_2_ + e^−^ →  *O_2_^−^ (−250 kJ/mol)Q + e^−^ →  *Q^−^ (−247 kJ/mol)Mg^2+^ + e^−^ →  *Mg^+^ (−69 kJ/mol)
*Radical-propagating phosphorylation*
ADP + Pi (+ *OH) →  ATP + H_2_O (+*OH) (36 kJ/mol)
*Radical-quenching phosphorylation*
ADP + P_i_ + 2*OH →  ATP + H_2_O + H_2_O_2_ (−150 kJ/mol)
*Other radical-quenching reactions*
*OH + *H →  H_2_O (−492 kJ/mol)*Q^−^ + e^−^ (+ 2H^+^) →  QH_2_ (−368 kJ/mol)*O_2_H/*O_2_^−^ + *H →  H_2_O_2_/HO_2_^−^ (−361/−322 kJ/mol)*O_2_H/*O_2_^−^ + *OH →  H_2_O/ OH^−^ + O_2_ (−250/−197 kJ/mol)*OH + *OH →  H_2_O_2_ (−186 kJ/mol)2*O_2_H →  H_2_O_2_ + O_2_ (−125 kJ/mol)
*Molecule-unbound ion–radical transmutations*
*H + O_2_ →  H^+^ + *O_2_^−^ (−211 kJ/mol)*Mg^+^ + O_2_ →  Mg^2+^ + *O_2_^−^ (−181 kJ/mol)*OH + H_2_O_2_ →  H_2_O + *O_2_^−^ + H^+^ (−100 kJ/mol)2*O_2_^−^ + 2H^+^ →  H_2_O_2_ + O_2_ (−176 kJ/mol)

### 4.4. Elaborate Comparison of ETC-CRAS and Murburn Bioenergetics

Even though we have brought out the fallibility of the ETC-CRAS model and highlighted the viability–tangibility of the murburn explanation via multiple modalities through many publications over the last 8 years, the researchers in the field remain glued to the acclaimed proton-centric idea, disregarding the crucial importance of oxygen. Therefore, to take things beyond reasonable doubt, we present irrefutable evidence/arguments via the three perspectives of: experimental findings (Table 1; which also includes the last six points as the observations misinterpreted to support ETC-CRAS!), structural features (Table 2) and theoretical considerations (Table 3).

Via the three tables above (comparing dozens of agendas), it can be seen that not in a single criterion of comparison (other than aesthetic disposition and the community’s inertia!) does the murburn model fare worse than ETC-CRAS, which should be considered a refuted hypothesis now.

### 4.5. Some Properties of Murburn Mechanisms

By being a stochastic mechanism [11,13,14,18], non-integral and variable kinetics is a hallmark of murburn reactions, as observed in physiology.ECS-coupled anionic DRS formation occurs at the redox centers such as *d* or pi-electron-containing cofactor systems like metals or flavins/retinoids (or both, like hemes) [1,4,18]. Thus, murburn concept is the primary rationale for electrical activity in cells. Even the activity of Na,K-ATPase is explained with DRS [19].In the murburn model, metabolic reactions work in parallel (deterministic series/chains are not mandated!) [4,16].They essentially abide by simple thermodynamic principles and pose minimal affinity-based requirements, offering a significant evolutionary advantage [1].As the presence of substrate exerts a thermodynamic pull, constitutional wastage of redox equivalents and collateral damage are minimized.The micro-dimensioned life forms were primordial owing to the necessity to curtail the free availability of protons for oxygenic systems to evolve subsequently.The evolution of Complex V is seen primarily as a pH-chemostat regulating murburn reactions at redox centers through proton supply, and an enhancer of ATP yields by serving as auxiliary murzyme (consuming H^+^; not recycling it!) at high respiratory rates [11].If DRS are produced in greater amounts, they could react among themselves, producing non-radical end-products and dissipating energy as heat (thermogenic uncoupling), an important inbuilt safety measure [20].Fast DRS-detoxifying enzymes in “solution” and e-buffering molecules like cytochrome *c* in aqueous milieus or ubiquinone in membrane milieus are yet other agents in the evolutionary repertoire that enable DRS-based physiology [16].Therefore, rather than seeing DRS as toxic and wasteful agents, it must be realized that DRS are the elixir of life. Yet, in spite of all the safeguarding measures (listed above) collateral reactions over prolonged periods can produce the deleterious effects of DRS, as chronically reflected in aging and acutely in pathophysiology.

Therefore, we arrive at the paradox: “*What giveth life also taketh life”!* Now, we stress yet another salient feature of cellular composition that originated very early: coacervation.

## 5. The First Cell Was (and Every Cell Now Is!) Essentially a Coacervate

By definition, a coacervate is a complex colloidal system with discrete phases. From simple mixtures of amino acids, microscopically visible coacervates (without a membrane, akin to membrane-less organelles) can be obtained resembling “*Fox microspheres*” [21]. Considering only the initially most abundant amino acids [3,22] with Pro, Glu, Asp and Gly unfolded peptides can be formed, which are sticky and promote formation of coacervates (microspheres), a huge step of increase in order occurs, from molecular to “cell” size, and from “normal” to “*polarized-oriented multi-layer*” or POM-water with completely different properties [22,23]. Insertion of the hydrophobic residues Val and Ala allows formation of more folded regions suiting primitive catalytic activities. Val and Ala can form the short precursors of stable α-helices [5,22]. Such coacervates acquire several typical properties of cells (the “instant” PCHEMS) although lacking DNA, RNA and ribosomes (essential ingredients of central dogma). Some salient features [1,5,9,22]:Accrete/adsorb additional oligomeric material (growth).Pinch off buds (physico-chemical basis of division).Produce POM-water exhibiting lower solvency for many solutes, including Na^+^, leading to exclusion without ion pumping, just a phase/phase equilibrium [6,22,23] together with DRS-related equilibriums [1].Water molecules in POM-water together with water-polarizing oligopeptides coherently behave as a unit [6], all vibrating at a common frequency [24,25,26,27], one of the most essential properties of life (as evident/experienced in the conscious state).Accumulate K^+^ (over Na^+^) even up to 1600 times [28] (without membrane pumps), just by selective adsorption on Asp and Glu carboxyl groups of the coacervate oligopeptide matrix [6,22,29], in conjunction with murburn equilibriums [1].Can swell and shrink by colloid osmosis, without a membrane [6,7,22]. Volume changes effectuate all physicochemical parameters, an overlooked metabolic regulatory mechanism assuring dynamic coherence (*cytotonus control)* [30]. Accordingly, actin-like proteins became the (overlooked) activator of oxidative phosphorylation [5,30], also influencing (overlooked) metabolic control by cell perfusion [31].An exclusion zone at their phase boundary characterized by charge separation and a boundary potential (without pumps), sensitive to environmental influences including IR/VIS/UV radiation [32] and internal-to-external DRS ratios [1], executes some sensory/effector functions. If prepared with added lecithin, the action potential trace obtained is almost indistinguishable from an axonal one [28]!Exhibit long lifetime despite being low-entropic, excited and coherent. Without long-term (meta-)stability, evolution is impossible [33,34].

## 6. Murburn and Coacervation

The properties listed above, along with Table 1, Table 2 and Table 3 illustrating murburn characteristics, mean that cells do not need to depend on an “intelligent membrane pumping ions” mechanism for deterministically governing their activities. Now, we can understand that cells take up energy to automatically organize in an equilibrium-driven fashion (although apparently far away from equilibriums in many distributions) to carry out their functions.

The DRS–coacervate cooperation demonstrates that life emerges as a logical consequence of the principles of chemico-physics and their inherent non-linear mathematical structure, which leads to an increase in complexity and coherence. Within these laws there is place for random processes, among which meta-stable realms and coherent solutions are intrinsically present [1,3,5,33,34,35,36]. The latter, if occurring, will be selected almost a priori on behalf of their immense advantages. This framework allows subsequent advantageous, workable functionalities to be integrated into the existing dynamic structure with already great stability, so that success can be continued and expanded. These non-linear properties clearly lead to workable and purposeful mechanisms and functions, as illustrated irrefutably above: “purpose” again enters evolutionary thinking as an inherent outcome of chemico-physical laws [33,37], and not due to overt vital determinism. The logic outlines of pre-biotic evolution presented above illustrate the early origin and inherent stability of PCHEMS obtained by just random chemistry, thanks to their embedding in the primary deterministic aspect of physical and chemical laws.

Epicenters of murburn activity would be expected to differ from other regions in phase behavior. Taking an overall thermodynamic approach (which includes entropic aspects), it could be interpreted that many complex physiological processes are characterized by “*order-to-disorder-to-order cycles*” for physiological work output which go along with phase transitions of the locally involved coacervate system. A most striking example is that of muscle relaxation-to-contraction-to-relaxation cycles. The new understanding of the bioenergetics of cells and tissue systems presented in this review (as discernible from Figure 2) integrates this perspective. It was first elaborated by Bauer in 1935 [26,38] (and agrees with the view of Schrödinger [10]), who saw “*work out of coherent order*” as the only thinkable alternative for the unreal mechanistic proposal based on the physics of diluted solutions [5,6,7,27,39]. The idea was further elaborated by Ling [40] and recently corroborated kinetically and topologically in living yeast cells [41,42,43]. However, murburn concept remained the missing and primary part within this picture, until its recent revelation and impact. The cellular system components are redox-sensitive (with DRS as effectuators) and the redox dynamics within locales also determine CEM profiles (not just ionic fluxes!). Action potentials do not only depend on Na^+^, K^+^, ATP and known transporters but more primarily on O_2_ and one-electron transfers generating DROS. The same holds for muscle contraction and other motility mechanisms. Missing in the latter investigations is the insight from murburn concept that DR(O)S at normal (low) physiological concentrations also bring about energizing/altering impacts in water–ion interactions and fluid lattice organization. It is already known that DROS can originate from oxidation of water with Fe^3+^ (Fe^2+^-UV/Fe^3+^ cycle: Box 1) and other suitable 1e-donors. Coherently polarized water can itself stock quasi-free electrons and execute “*water respiration*”, i.e., can act as a quite strong electron donor [26], and that DROS can directly effectuate solute–ion–water equilibriums [1], indicating a direct cooperation between coacervation and murburn physical chemistry (Figure 2 and Figure 3). As a next step, early DROS-dependent ATP-synthesis needs low water activity, which typically exists in coacervates, thereby connecting murburn and coacervation.

## 7. Electromagnetic Causes/Effects of Murburn–Coacervation

Stochastic murburn reactions effectuate the “primary” powering of chemical, electrical and mechanical processes. They also bring a kind of order and coherence (networking of components) by ensuring the most basic level of homeostasis through the logic of interconnectedness of all equilibriums inside (in contact with those outside) and maintenance of solvent/metabolite concentrations (owing to solvency, adsorption and osmotic/turgor forces), altogether making up an “analog/digital” sensory–integrator–effector system [1,14]. Besides the primary energizing through DRS-ECS in respiration [1,4], photosynthesis [1,44], photoreception [1,18], photodynamic therapy [44], etc., cells also possess a “secondary” cellular energizing system derived from the primary murburn dynamics (Figure 2 and Figure 3) via:(i)ATP adsorption/hydrolysis indirectly and intricately combined with direct DRS effects on “*order disorder cycles*”; and(ii)electromagnetic (EM) emission followed by re-absorption (appreciated by physicists but neglected by most biologists [5,24,25,26,27,45,46,47,48,49,50,51,52,53,54,55,56,57,58,59,60,61]).

According to Maxwell’s laws, moving charges produce an EM field, i.e., displacements of electrons (localized excitations, e-transfers, Lewis effects, etc.) and mineral, organic, macromolecular ions and larger charged structures (often pulsed group processes at longer wavelengths). In cells, it spans the range from a few Hz to the photonic range (IR/VIS/UV). It is amply demonstrated that many frequencies are internally re-absorbed by means of resonance and are used to coordinate cellular activities [46,47,48,49,50,51,52,53,54,55,56,57,58,59]. Herein, the exergonic murburn processes/reactions are the primary emitters and may produce “high-quality energy” [17] radiation (even UV! Box 3) which other cellular systems capitalize on through re-absorption. Therefore, cellular murburn equilibriums give not just interactions among molecules and ions and are not just be affected by radiations [18,44], they also result in the creation of localized electrode effects and complex macroscopic electromagnetic fields forming a CEM [45], whose effects can be affected/compounded by coacervation.

In coacervates (but not in dilute solutions) compounds are ordered so that the produced EM field can reach a certain degree of coherence, particularly since most water molecules are also ordered [5,6,7,24,25,26,27]. Though murburn reactions are stochastic, they take place in an overall milieu which is highly ordered (particularly water). The DRS-dependent *in-fluid* ATP synthesis occurs with ordered water of low activity [4]. Though the direct EM emissions by murburn reactions may have limited order, surrounding coherent water or other ordered arrangements of molecules can take up the radiation and re-emit it in coherent form, in turn resonantly reaching other ordered systems (microtubules, centrosomes, centrioles [48,49,50,51,52,53,54], DNA [55,56]) with a broad absorption range. This would allow conversion into coherent EM waves, re-emitted at (different) typical frequencies, which may be instructive to diverse cellular activities [5,45,46,47,48,49,50,51,52,53,54,55,56,57,58,59,60,61]. In present-day cells, the degree of coherence is high; for microtubules even quantum coherence and photonic resonant “*phase-correlated*” signal transduction are directly observed [51,52,53]. For instance, cytoskeleton-associated EM-field dynamics deep in cytoplasm precede and orchestrate ionic effects at the neuron membrane [53]! There is much evidence that the cellular EM matrix could serve as a cue and can even directly integrate chemical [46,47,48,54,59], electrical [51,52,53] and mechanical [54] activities, morphogenesis, tissue repair [57,58], brain functioning and development [51,52,53,54], tissue-wide DNA intertalk [55,56] and information transfer between separated individual systems [60,61]. The remarkable “mitogenic” UV information transfer from a germinating onion to a non-germinating onion, discovered by Gurwitsch in 1912 [5,17,46,57,60], made him suggest that high-energy DRS reactions are responsible—they feature since the origin (Box 3) and may be the only reactions able to reach UV emission. The overall bioenergetics scheme of interactive/holistic cellular networking and energy flow is captured in Figure 3.

## 8. DRS–Coacervate Dynamics, CEM, Cancer and Degenerative Diseases

It is now established that ECS-DRS-based murburn mechanisms are involved in all types of cells at ubiquitous loci [1], thereby serving as a metabolic/physiological pillar of life. It is thus reasonable to posit that “healthy” murburn-DRS levels/dynamics are associated with coordinated electromagnetic patterns throughout the various protein/organellar/cellular/tissue loci (Figure 2), which could also depend upon local functions and topologies. Sonnenschein and Soto [62] regard “*proliferation with variation and motility*” as the “*normal state*” of all cells and the state of cells in a healthy well-organized, differentiated and well-controlled tissue as the “*special state*”, obtained as the result of contact inhibition with a whole set of regulatory mechanisms connected to it. Interactional problems among neighboring cell types may induce development of cancer, whereby cancer cells regain the “*normal*” state of “proliferation with variation and motility”. An important test validating this hypothesis stems from the observations that: (a) cancerous and degenerated cells show distinct morphologies, water structure, coacervation properties and EM field as compared to healthy cells and (b) cancer cells can reverse their malignant state when experimentally placed in a sufficiently extensive healthy tissue environment [63,64,65].

External measurements of the EM field show only the small amount of EM radiation produced in cells, tissues and entire bodies, which has not been internally re-absorbed [46]. Yet it is an indication of what happens inside. Of particular importance is that normal cells decrease their emissions with higher cell density, which means a higher degree of energy recuperation. In contrast, tumor cells have increased emission with higher cell density, and this becomes more pronounced the more malignant the cells are! Also, photonic emissions (“biophotons”) of tumor cells lose their coherence [46]. This is correlated [5] with observations of Damadian [66] that cell water ordering typical of healthy tissue is diminished or lost in tumors, as directly shown by proton-NMR (and MRI development thereof). Fröhlich [24] derived the coherent aspect characterized by its common frequency for the excited polarized protein–water system. Thereafter, he developed the idea of the existence of a global coherent oscillation with a characteristic long wavelength compared to the size of cells, whereby “*phase correlation*” over the entire tissue *synchronizes* its mechanical–electrical dynamics [67]. Destabilization of this unifying system occurs when a critical number of cells lose the common frequency, below and above which the destabilized cells respectively recover the right frequency or become disordered, with altered mitotic and adhesive properties. Webb [68] experimentally corroborated it by showing that healthy tissue emits only two series of coherent millimeter waves, one associated with proteins, the other with DNA, whereas tumors emit greater quantities of lower-frequency, less coherent radiation. This implicitly suggests the possibility of reversing the cancerous state (as observed [63,64,65]), if the interactions/signatures could be fixed with chosen frequency inputs methods [69]. We envisage a time when, just like MRI is used for assessing the structural states of tissues, functional DRS-EM dynamics could be addressed with such investigative, but also therapeutic, approaches.

Warburg [70] argued that an increased glycolytic–respiratory ratio is “*the*” cause of cancer, which was statistically confirmed by Acebo [71]. Why do well-oxygenated tumors convert glucose to lactate in cytoplasm instead of subjecting it to mitochondrial respiration? Is respiration suppressed (as many believe), or is there excess DROS production through mitochondrial overload, another possibility to explain the Warburg effect [72]? The latter agrees with decreased water ordering [56] and with murburn concept, whereby overload induces radical activity/quenching, which may produce excessive “mitogenic effects” [60]. These imply that factors messing with mitochondrial function can be causative, including radiation of diverse wavelengths (from X-rays to even ELF waves), toxic metal ions, pesticides, components of some cosmetic and household products, hydrogenated fats and additives in processed food, soot particles, air pollutants including DRS, etc. In a meta-study covering over 2000 articles, Geesink and Schmieke [73] distilled the mathematical “*frequency law*” determining which frequencies are dissonant with the body (harmful, destabilizing) and which are harmonic (beneficial). Such approaches/data could aid biomedical and other device developers in the future! Plankar et al. [74] collected the arguments which demonstrate that *loss of coherence precedes mutagenesis*, supporting Fröhlich’s hypothesis. Our works suggest that healthy mitochondrial function is a crucial cellular ECS-DRS-EM-coherence contributor, which, if impaired, is perhaps responsible for pathogenic states. Such murburn-based insights also allowed us to propose new perspectives to address cancer therapy [75].

## 9. Summative Ideas

Herein, we brought to light the following key points

Murburn concept constitutes a paradigm shift in bioenergetics.It explains why aerobic life needs oxygen for instant functioning.The physiological roles of DRS at low concentrations are disclosed.Cooperation between coacervation and murburn mechanisms is indicated.Dynamics of DRS are crucial for life’s origin, evolution and (patho)physiology.Murburn insights suggest new approaches to aging and therapy.

Further elaborating:**A**.Murburn concept (ECS-DRS) forms the primary drive and bioenergetics pillar of cellular life. From this perspective, it can be easily discerned that TMP is a consequential effect of the ECS-DRS murburn process and not the driving force of life. The useful work performed by an automobile is not directly from exhaust or noise (which is analogous to the TMP), but it is because of the “burning of fuel which produces an expansive gas” that the vehicle runs! Murburn is the core logic of how cells automate and integrate to function as “simple chemical engines” (SCEs), a thermodynamically viable system that does useful work, and how “electronic intelligence emerges” (and is not resident in the ion-pumping membrane, as misperceived earlier!) from a discretized distribution and network of molecules and ions.**B**.The pre-biotic “burning” activity of iron producing DR(O)S initiated the direction towards life: synthesis of hydroxy-, keto-, carboxylic and amino acids and later ATP by random murburn chemistry. Early coacervations increased complexity from the molecule to the “cell-size” level and from “normal” water to highly ordered water, affording several typical life properties (except those of the central dogma). A significant amount of the emitted electromagnetic energy is re-absorbed by coherent cellular systems (microtubules, centrosomes, centrioles, DNA, etc.) and re-emitted coherently at different wavelengths, which is “healthy dynamics” for the normal functioning of cells. External radiations and select chemicals could perturb DRS metabolism, leading to a disturbed (less coherent) CEM affecting cellular coordination, including chromosome/DNA integrity, explaining why cancer/mutagenesis is preceded by decreased coherence [74]. The non-genetic/hormonal/neuronal modalities of regulations/control via analog–digital logics [14,76] should be investigated in this context. While much remains to be explored, we have conclusively demonstrated that ideas like ETC-CRAS and ion pumping do not abide by the laws of physics, whereas the murburn–coacervate view (and associated CEM view) affords us a tangible/holistic perspective to understand the underlying chemical–physical logic of health and diseases. This was elaborated upon in the two-part review of how murburn concept enables spontaneous and automated operation of multiple systems, starting from a molecular level and seamlessly integrated all the way to the macroscopic levels [77,78]. Many recent investigations demonstrate that DROS play an important, though not understood, adaptive, regulatory role in cell motility [79,80,81,82,83]. DRS (murburn) can also bring about inheritable and epigenetic changes in DNA (and also cross-react with diverse cellular components) and its cumulative effects lead to death. Thus, murburn can drive evolution, generate diversities in species and also explain differences among identical twins.**C**.Chemico-physical mechanisms in (biological) living systems are affected and explored/ratified by probing with inhibitory agents. Murburn concept is the only thermodynamic/kinetic explanation available explaining the globally debilitating effect of cyanide on physiology, ascertaining the significance of DRS to life, with full context to the structures of proteins and membrane physiology [84,85,86,87]. While the primary oxygen activation reaction is: O_2_ + e^−^ → *O_2_^−^ (−250 kJ/mol) (which explains how the mitochondrial/chloroplast quinones can effectively enhance catalysis via this equilibrium, as its one-electron reduction is comparable with −247 kJ/mol), the cyanide-catalyzed DRS cycling to water is: *OH + *O_2_^−^ → OH^−^ + O_2_ (−275 kJ/mol) [87,88]. Thereby, when cyanide is present, DRS are unavailable for essential life-sustaining activities owing to electrons sinking into the life solvent of water, as espoused by the murburn “thermodynamic pull” theory. The stoichiometric heme binding at cytochrome oxidase fails as an explanation in this regard [86]. Further, it is improbable that cells/neurons also evolved highly specific high-throughput ion channels as some currently believe. If so, how could cyanide perturb the cellular functions? Since (i) DRS like NO (and CO too!) are recognized as a molecular messenger [89] and DRS-based signaling is described more and more in the literature [90], (ii) DRS production is also known to be directly correlated to ATP synthesis (and trans-membrane potential) in mitochondria in routine physiology [91] and photodynamic (low-level laser) therapy [92], (iii) DRS production activities are associated with good health measures like exercise [93] and longevity [94], (iv) the perception of DRS is changing [95], and redox activities of small molecules like vitamin C and proteins like cytochrome *c* have been recharted, our attribution to murburn concept as a founding bioenergetics–coherence principle and seamless integrator of chemico-physical functions of life [1,14,76,77] is validated. Affording such recognition also enables us to specifically explain diverse conundrums like: non-specific post-translational modifications, the unusual hormetic enhancement of heme-enzyme activity by azide (erstwhile presumed to serve as a potent inhibitor!) and idiosyncratic effects of diverse drugs, lack of stereoselectivity in biological halogenation reactions [1,77,78], etc. Recently, researchers found that mitochondria isolated from cancer or non-cancer cells could impact the respiratory activity of each other when they were physically separated from one another [96]. Such non-chemical signaling between disconnected mitochondria can only be explained via electromagnetic radiations, which can be produced by the high-energy-yielding DRS/ROS reactions (some of which are listed in Box 3). Thus, such potent and bewildering findings can also be efficiently reasoned with murburn concept! Quite simply, with murburn concept, we can understand the Janus, i.e., both the Dr. Jekyll and Mr. Hyde, persona of DRS/ROS, reported in diverse contexts [97,98,99]! Slowly and surely, the murburn bioenergetics principles are being cited, discussed and accepted in scientific community [100,101,102,103,104,105,106,107,108], and some textbooks of cell biology [109] and medical biochemistry [110] have incorporated it. It is now imperative to recognize that stochastic intermolecular interactions (and electromagnetic radiations/fields thereof) and collective array of interconnected events mediated via DRS/ROS also contribute towards sustaining life, as is being explored and unraveled constantly [111]! Table 4 presents an appraisal of select systems where we have convincingly demonstrated the applicability of murburn concept.

**D**.Supposing a murburn reaction releases ~500 kJ/mol (is exergonic!), the outcome could emit: 1–2 UV/Vis photons by electronic transitions (if the states are excited), decades of IR photons by vibrational modes (perhaps the most dominant outcome!), thousands of microwave/radiowave photons (if spins or dipoles are involved, as in radical recombination and ionic current oscillations) and trillions of extremely low-frequency (ELF) quanta (if large-scale macroscopic charge mobilization occurs). The exact distribution depends on the reaction mechanism and the microenvironment (whether in water or at lipid interface). In most cases, IR dominates (explaining the higher temperature of living beings), but specialized reactions can shift into other bands, depending upon the nature of molecules involved. Table 5 presents the panoramic spectrum of interactions that the murburn paradigm could help understand, given that only murburn reactions could emanate higher energy frequency radiations. If we consider that the energy of breaking an O-O bond in a peroxide molecule is around 142–146 kJ/mol (~820 nm photon), then the explanation for photodynamic therapy (observation of increased ROS and ATP production, alteration of mitochondrial TMP, etc.) is easily afforded with the murburn theorization that the hydroxyl radical formed leads to the outcomes. Therefore, it is not anything perverse to imagine that reactions that could give high energy yields can also emit various spectral bandwidth radiations (as shown in Table 5), and all these may bring about an intermolecular or inter-structural coherence that was not deemed possible earlier. Cell biologists and medical professionals should open up their minds to these new possibilities. Most importantly, as seen from all contemporary textbooks, the erstwhile bioenergetic explanations for cellular respiration (or photosynthesis or thermogenesis) afforded low free energy yields in their steps (examples: the highest-energy-yielding reactions of glycolysis or the Krebs cycle provide lower energy than what is afforded by ATP hydrolysis). This write-up provides a tangible connection of the stochastic ECS-DRS fulcrum of life with the impact and/or generation of radiations in (patho)physiology, of both low and high frequencies. Therefore, murburn concept constitutes a holistic perspective of matter–radiation interactions in/among living and ecological systems.

**E**.The murburn principle predates genetic mechanisms as a foundation of life; therefore, we can clearly envision how crucial it is for understanding life! In spite of having identical genes, the operation of murburn in cells induces chance-based differences between siblings; becoming a primary cause of idiosyncrasies. The classical perspectives in epigenetic and post-translational modifications are purely active-site enzyme-centered highly specific reactions. Murburn insights extend it to have relevance even outside the active site [97], independent of S-adenosyl methionine and ATP. Murburn concept can also help us reason that nucleic acid materials went on to incorporate phosphates and pentose sugar with hydroxyl groups within them to enhance the ruggedness of the molecular sequence to external radical attacks. Murburn reactions in CYP enzymes can generate genotoxic intermediates from procarcinogens. Excessive ROS (due to murburn activity in mitochondria or cytochrome P450 systems) can lead to oxidative stress, causing aberrant epigenetic changes linked to cancer initiation and progression. Increased ROS from murburn activity can mutate DNA (particularly in mitochondria!), activate oncogenes or silence tumor suppressors via epigenetic mechanisms and even affect activities of epigenetic enzymes. Cancer cells often exhibit the Warburg effect (aerobic glycolysis) and mitochondrial dysfunction, which may involve murburn-type redox processes [75]. Murburn concept provides a fresh perspective on how redox reactions influence epigenetics and cancer, particularly through ROS-mediated epigenetic alterations and metabolic rewiring. Posing immense application potentials (refer to Table 4 for a listing of systems already unraveled) in diverse fields and offering a true possibility of understanding aging and several physiological and pathological mechanisms, DRS dynamics and murburn concept are a treasure trove waiting to be opened [115,116,117]! Further research could uncover novel biomarkers or therapies targeting murburn pathways in routine physiology/oncology and mainstream traditional Chinese/Indian ways of medicine.

## 10. Outstanding Concerns and Questions

Understanding any/most instantaneous molecular to macroscopic transitions (say, a photo-transduction event, or a muscle twitch or a neuron firing) would necessitate us to critically indulge several entrenched/redundant ideas/terms like proton pump, *pmf* & chemiosmosis [118], rotary motor-protein functions in ATP-synthesis [119] & flagellar motility [120], ETC [121] & Z-scheme [122], Kok-Joliot [123] & Quinone [124] cycles, high selectivity/flux cation pumps [125] & channels [126], etc. and have thermodynamically/kinetically justified murburn models and mechanisms in place. Paradigm inertia (reluctance of systems to change tracks), inadequate exposure or tracking of pertinent literature, established funding/publication practices, propensity for misinterpretations and aesthetic dispositions (seeing DRS as only a toxic or wasteful and unavoidable aspect of life) and the “questionable moral disposition” of some individuals are the major detriment to ushering in better health, longevity and quality of life prospects (sponsored by “murburn awareness”) for our forthcoming generations. If we have more informed and open-minded scientists, we would proceed to addressing problems such as:Temporal and spatial landscape of DRS dynamics around proteins.DRS–protein (surface)–water, DRS–coacervate and DRS–EM interactions.Magnitudes and variations of various EM radiations/fields.New thermodynamic foundations on matter–energy exchange in cells.Dynamic controls determining various types of chemo/phototaxis and motility.DRS-mediated overall digital/analog logic of regulations.DRS-mediated one-electron outcomes in cybernetics.

## Figures and Tables

**Figure 1 ijms-26-07542-f001:**
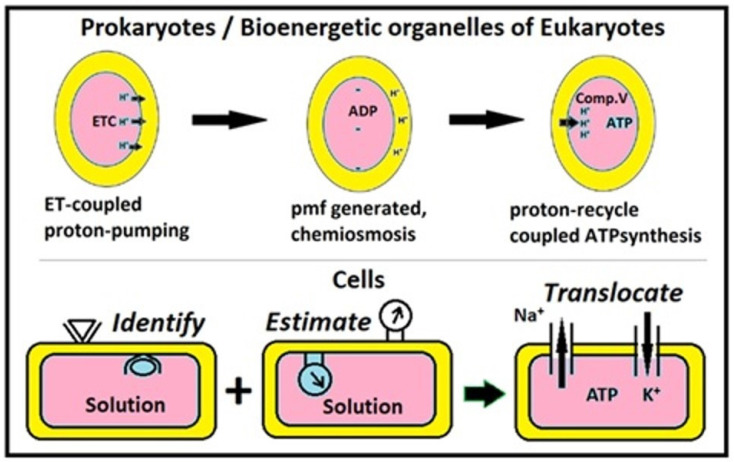
Powering and homeostasis from the ETC-CRAS perspectives. The classical/acclaimed perspective treats cells as dilute solutions and cell/organelle membranes as intelligent guardians that pump/permit ions to generate/dissipate powering gradients [1].

**Figure 2 ijms-26-07542-f002:**
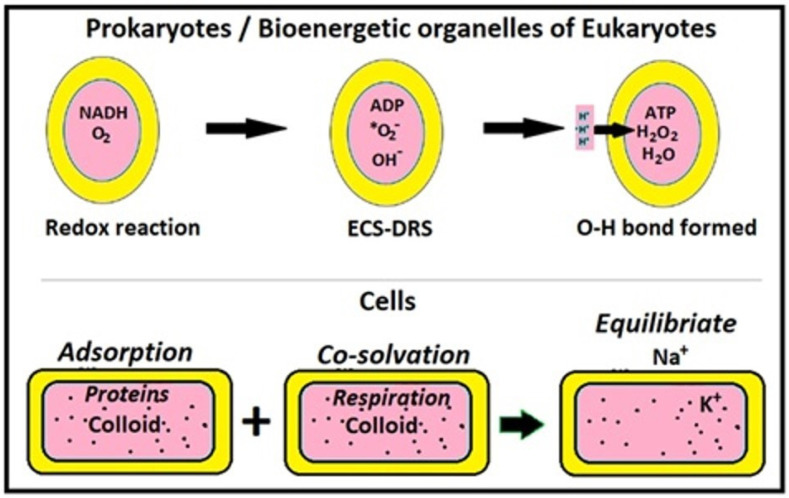
The murburn bioenergetic view deems cells as discretely packed fluidic colloids that are powered ultimately by simple redox reactions. On the way to forming water, oxygen gets transiently reduced to superoxide (*O_2_^−^) and hydroxide (OH^−^) ions, which gives the negative TMP. This view abides by the laws of physics whereas the ETC-CRAS view is thermodynamically untenable [1,4,13].

**Figure 3 ijms-26-07542-f003:**
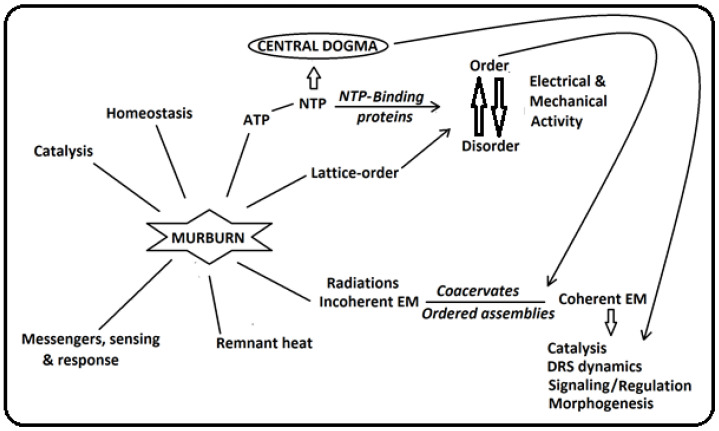
New understanding of real-time cellular energy flow/bioenergetic and work output start at murburn reactions, which: (a) not just effectuate fluid homeostasis, catalysis, sensing–responses, but also produce/influence lattice order directly and indirectly through murburn synthesis of ATP (NTP); the above are involved in order–disorder cycles in the form of coacervate phase transitions executing electrical and mechanical work and (b) through EM radiation, a significant part gets transformed from incoherent to coherent by ordered macromolecular assemblies and serves to coordinate cellular functions at acute time scales (molecular signaling via central dogma is too slow but can consolidate this fast coordination at larger time scales). Some amount of energy is radiated to the environment largely as remnant heat. Produced NTP is also necessary for long-time central dogma aspects.

**Table 1 ijms-26-07542-t001:** Experimental observations and explanations in bioenergetics (especially in mitochondria).

No.	Criterion/Observation	ETC + CRAS (X)	Murburn (Y)	Analytical Comments
1	*Oxygen’s role and formation of DROS in mitochondria and chloroplasts*	Final e-acceptor; stays bound at Complex IV to form water; DRS seen as wasteful/toxic byproduct.	O_2_ forms DRS which interact with ADP/Pi to make ATP; DRS are essential intermediates.	Y (not X) gives direct redox phosphorylation. Y (not X) explains DRS production.
2	*Proton gradient (ΔpH) and membrane potential (Δψ); bioenergetic membrane must be intact*	ETC’s H^+^ pumping must generate ΔpH of 3 units (1000-fold H^+^ gradient or Δψ of ~ −200 mV). Cannot explain how ATP synthesis goes on with permeabilized or leaky membrane.	Small ΔpH/Δψ is an epiphenomenon due to ECS-DRS. Not *pmf*, but ECS-DRS mechanism by O_2_ and murzymes forms the phosphorylation drive. Practically aprotic environment makes O_2_-centered DRS more stable. Intact membrane prevents loss of DRS; but is not a must!	No protons to pump in micro-dimensioned organelles! (How can proteins be selective for 0.02 pm H^+^?); and little pH gradient present! ATP synthesis observed with DRS (even in systems sans ΔpH or Δψ). DRS directly correlates with ΔpH and Δψ in closed organelles. Supports Y, discounts X.
3	*Stoichiometry (component:ATP ratios; H^+^:P, NADH:P, O:P)*	Requires tight stoichiometry: Originally 6 ATP per NADH or per O_2_. (Modified to <5 later.)	Loose coupling via stochastic interactions of ROS with ADP/Pi near inner membrane Complexes I to V. Ratios can be fractional due to competing reactions.	Data from diverse labs show variable ratios (even >6). Data spectrum supports Y (not X).
4	*Heat generation in brown fat by thermogenin (UCP)*	Uncoupling proteins dissipate *pmf* by tranlocating protons with fatty acids to give heat. (Shaky physics?) Requires fatty acids.	Outward translocation of DRS or inward proton movement facilitated by lys-arg of UCP gives water and heat at interface. Does not require fatty acids.	Structure of UCP and fundamental thermodynamic considerations support Y (not X).
5	*Enhancement of ATP synthesis by xanthine/xanthine oxidase (XO)*	Fails to explain the phenomenon for mitochondria and chloroplasts.	Xanthine/XO gives superoxide, which aids ATP synthesis.	Supports Y, not X.
6	*No DRS, no ATP! Provision of higher NADH/(ADP+Pi)/O_2_ increases both DRS and ATP in milieu.*	Cannot explain.	Organellar system works to make ATP via the intermediacy of DRS.	Supports Y, not X.
7	*Provision of NADH progressively reduces Comp I, III and IV (Kaelin)*	ETC concept! Electrons serially relayed in ETC from Comp I by CoQ (to Comp III) and Cyt. c (to Comp IV), while O2 stays stuck at Comp IV.	NADH reaction with Comp I produces DRS, which reduce CoQ in membrane, Cyt. c in milieu and form peroxide. All these reduce Comp III and Comp IV in parallel ways.	Q-cycle is impossible. Oxygen does not and cannot stay bound to Comp IV. These discount X and known features support Y.
8	*ATP formation with induced pH gradient (sans light) in chloroplasts. (Jaegendorf–Uribe)*	pH gradient drives ATP synthesis in bioenergetic organelles by H^+^ cycling!	External H^+^ as a reactant drives many photosynthetic reactions to the right. Supports the murburn thermodynamic role of H^+^ consumption in the bioenergetic process.	Three-unit pH gradient is non-physiological (unseen in chloroplasts!). The experiment was ill-constructed and the observation was misinterpreted in favor of X.
9	*Light-induced synthesis of ATP in vesicles with membrane-embedded Comp V + bacteriorhodopsin (Racker–Stoeckenius)*	Light triggers rhodopsin to pump protons out from vesicles, which return via Comp V to make ATP inside.	Rhodopsin Schiff’s base pK_a_ > 13; cannot pump protons but is a photoactive DRS generator, and the DRS makes ATP.	The experimental control of rhodopsin alone gave 20% ATP formation with light. Does not agree with X, but supports Y.
10	*Effect of uncouplers like dinitrophenol or DNP (Mitchell)*	Disrupts *pmf* needed for ATP synthesis. So oxyen used wastefully; uncoupler shuttles protons inward.	DNP impacts interfacial DRS dynamics and makes water.	If the minuscule protons cannot move across the membrane, how can DNP (which is more massive and has more charges!) flip-flop? Mitchellian explanation fails! Findings support Y, not X.
11	*Effect of oligomycin binding to Comp V (Lardy–Racker)*	Blocks F_0_ and stops ATP synthesis even with Δψ; because protons cannot come in via Comp V’s F_o_.	F_o_ of Comp V serves as an ATPase and pH stat. F_1_ of Comp V serves as a proton-consuming ATP synthase.	Findings misinterpreted in favor of X; observation actually supports Y.
12	*Effect of rotenone, antimycin B and cyanide (CN) on Comps I, III and IV (respectively)*	Bind to Comps I, III and IV, respectively, to disrupt proton pumping by ETC, thus lowering ATP synthesis by definite and proportionate ratios.	The toxins disrupt DRS dynamics in murzone; either preventing DRS formation and interaction or subverting DRS to form water. DRS quenching by CN is very facile.	O_2_ can outcompete CN to bind Comp IV and toxic dosage of CN is too low (orders lower than hemeFe). Inhibition is too fast to be explained by binding mechanisms. Negates X; fully supports Y.

**Table 2 ijms-26-07542-t002:** The two models’ views on structure–distribution of molecules and architecture of cell/organelle.

No.	Component(s)	ETC + CRAS (X)	Murburn (Y)	Analytical Comments
1	*NADH and O_2_*	No particular significance attributed to structure of NADH; O_2_ binds and reacts only at Comp IV.	2e, 1H in closed aprotic ambiance allows O-centered 1e chemistry (H_2_O_2_ and H_2_O formation). Mobile O_2_/DRS interact with all components!	In the reducing ambiance, the highly mobile O_2_ cannot stay stuck to Comp IV. Indeed, the formation of DRS has been documented for Comp I through Comp IV. Discounts X; supports Y.
2	*Comp I*	Collects electrons from NADH and relays to CoQ, pumping out H^+^ in the process.	Receives electrons from NADH, binds ADP and generates DRS in the murzone for direct phosphorylation. Many FeS centers in the matrix-subtending arm (with many ADP-binding sites) make it a redox ADP-phosphorylating murzyme.	Some Fe-S non-route and some of unfavorable potential. ET along FeS in the matrix arm cannot be spatio-temporally coupled to any H^+^ pumping by membrane foot. CoQ cannot squirm into the protein to receive electrons. Discounts X; clearly supports Y.
3	*Comp II*	Collects electrons from succinate to pass on to CoQ in the membrane. Cannot explain the presence of multiple redox cofactors.	Highly localized DRS with FAD; a membrane-embedded redox phosphorylating murzyme.	Hardly any purpose for this enzyme with X! Presence of multiple redox cofactors (flavin, FeS, heme) and an ADP site supports Y.
4	*Comp III*	A complex enzyme that binds many molecules at the same time to convert CoQH_2_ to CoQ, reduce Cyt. *c* and pump H^+^ out.	A murzyme that recycles electrons lost in the membrane phase by making DRS from CoQH_2_ and enabling bound ADP’s phosphorylation.	Structure, kinetics and thermodynamics do not support X. Simultaneous bindings of many molecules at the lipid phase is not solicited by Y. The solvent-accessible heme and ADP-binding site on apoprotein support Y.
5	*Comp IV*	Oxidase binding O_2_ to receive 4 electrons from Cyt. *c* and availing 4 H^+^ to make 2 H_2_O molecules to pump out H^+^.	The peroxides formed in phosphorylation and DRS reactions are completely oxidized to water, making ATP in the process.	The terminal e-acceptor role attributed to O_2_ is a “meaningless role” in X. Peroxidative roles of Comp IV are known, supporting Y.
6	*Comp V*	Reversible rotary enzyme that dynamically harvests proton gradient via electro-mechanical forces to make ATP.	An ATPase at F_o_ (chemostat) and a DRS/H^+^-dependent ATP synthase at F_1_ (using incoming H^+^ as a reactant).	The presence of multiple ADP sites on F_1_ of Comp V discounts Boyer model and the fact that pure Comp V is an ATPase negates X. The above facts and in vitro H^+^-driven ATP synthesis support Y.
7	*Respirasomes*	Cannot explain.	Structures evolving to minimize murzone and DRS wastage.	Support Y (not X).
8	*CoQ*	An “intelligent” and mobile e-relay agent, ferrying electrons and protons from Comp I and Comp II to Comp III, where it has both e-donating at accepting roles.	E-sponging roles in membrane to prevent the escape and loss of DRS; different lengths to enable better stacking in the lipid.	Dual role at Comp III unviable in X. The larger CoQ_10_ (relatively immobile) is abundant, discounting transport role attributed in X; Quinones’ diversity supports the e-sponging role advocated in Y.
9	*Cyt. c*	“Intelligent” mobile electron-relaying agent from Comp III to Comp IV in the intermembrane space.	E-sponging roles in matrix and intermembrane space. Heme’s solvent accessibility supports murburn model.	Cyt. *c* is present in both phases of mitochondria. Initiation of apoptosis with the leaching of Cyt. *c* is an evolutionary strategy justifying Y.
10	*Inner membrane*	Impermeable to H^+^ and intelligently regulates pmf. Seeks placement of ETC in specific order and ratios on the cristae.	Packed with redox proteins to enhance ECS-DRS at the interphase. Cardiolipin increases anionic density, preventing DRS escape.	X does not explain preponderance of cardiolipin.
11	*Outer membrane*	Needed to form an intermembrane space capable of containing *pmf*.	A second membrane is present to minimize DRS escape. (No specialty required and none present!)	The fact that intermembrane space and periplasm equilibrate with cytoplasm negates X.
12	*Overall architecture and distribution*	The deterministic model seeks the orchestration of a highly complex mechanism across three phases, with hundreds of gene products, all for making one molecule of ATP!	The stochastic model works as a soup of chemicals, in various architectures and arrangements. Each component can work independently; and the system could easily evolve incrementally for the optimization of ECS-DRS mechanism. (Murburn model viable in erythrocytes, which lack mitochondria!)	X is discounted as it does not address the micro-dimensioned aprotic nature of bioenergetic organelles and bacterial cells. The diverse types of architectures found in mitochondria and the scattered and randomized distribution of proteins therein support Y. DRS-mediated ATP synthesis shown in vitro and in vivo. X requires too many components and unheard of modalities!

**Table 3 ijms-26-07542-t003:** Theoretical overviews of the two bioenergetics models.

No.	Aspect	ETC + CRAS (X)	Murburn (Y)	Analytical Comments
1	*Thermodynamics*	Endergonic.	Exergonic.	Supports Y, negates X.
2	*Kinetics (e.g., for Comp IV: unusual aspects like K_M_ < K_d_; anaerobic ET orders slower than aerobic ET)*	Slow, binding-based; anomalous kinetics cannot be explained.	Fast, collision-based; DRS-based mechanism explains atypical kinetics.	Supports Y, negates X.
3	*Pathway/steps*	Serial + sequential.	Parallel + unordered.	Y more probable.
4	*Coupling*	Indirect, electro-mechanical to chemical (?)	Direct redox phosphorylation; a simple chemical reaction; explains Cohn’s data of fast/multiple labeled O-atoms’ insertion in ATP.	X is unheard of elsewhere! Reaction logic supports Y.
5	*Molecularity of intermediates*	Multimolecular (e.g., Q-cycle).	Bimolecular (or trimolecular).	Y is more probable.
6	*Stoichiometry*	Fixed and whole number due to binding-based logic.	Variable and non-integral due to uncertainty of DRS function.	Data supports Y. X negated.
7	*Binding of molecules to proteins*	High-affinity interactions solicited with long-lived intermediates.	Not fastidious about binding affinity, as the reaction is delocalized.	Data supports Y, as diverse molecules donate and receive electrons to/from the system.
8	*Reversibility*	Yes.	No.	Y is more sensible than X.
9	*Conformation change needed?*	Obligatory.	Optional.	Y more likely than X.
10	*Long-distance ET at unfavorable redox potential*	Required!	This does not seek such seemingly improbable events.	X is more unlikely and unsupported than Y. Promiscuity of ET supports Y.
11	*Schema*	Deterministic.	Stochastic.	Y is more likely than X.
12	*Probability*	“Irreducibly complex”.	Simple, tangible, viable and probable.	Ockham’s razor prefers Y over X.

**Table 4 ijms-26-07542-t004:** Systems or fields with convincing models based on murburn concept (via several publications).

No.	System/Field	Classical Model	Advantage(s) of Murburn Model	References *
1	Peroxidase/Catalase (peroxisomes, ecological cycling of lignocellulosics and halogenated organics)	BROS-Compound I	Explains substrate diversity and inhibition; modulation by diverse molecules, practically diffusion-limited catalysis.	[1,4,14,77,84,85,87]
2	Cyclooxygenase & antibody function (inflammation; immunology)	BROS-Compound I	Explains substrate diversity and variable products; modulation by diverse molecules.	[1,111]
3	Lactate dehydrogenase (how the reverse reaction is viable for lactate to pyruvate!)	Reversible function	Reasons why lactate has to go into mitochondria or be taken to liver to use the energy equivalents.	[1,75,77]
4	Hemoglobin structure-function correlation (erythrocyte viability)	No equivalent!	Proposes the first rationale for prolonged viability of erythrocytes in the absence of nucleus and mitochondria.	[1,77,111]
5	Drug/Xenobiotic metabolism (how cells deal with “chemically unknown” molecules)	P450cam	Explains diversity of CYPs (cytochrome P450), promiscuity of CPR (CYP reductase) and Cyt. b_5_ and diversity of substrates.	[1,14,15,77,86]
6	Mitochondrial respiration (ATP synthesis, cyanide toxicity & thermogenesis)	ETC + CRAS	Thermodynamically viable rationale for oxidative phosphorylation and heat production in cells. Primary drive for life!	[1,4,5,11,13,14,15,16,20,77,78,84,85,86,87,88]
7	Light reaction of photosynthesis (radiation to reaction logic)	Z-scheme and Kok–Joliot cycle + CRAS	Discrediting the Z-scheme model, explained the roles of photosystems and antenna complexes.	[1,11,16,18,44,77,78,86]
8	Retinal phototransduction (radiation to electrical signal)	Retinal cycle (11cis to all-trans model)	Details direct role for oxygen in photoreception, explaining eye structure.	[1,18,78,86]
9	Na^+^/K^+^ ion-differential and electrosignaling physiology	Classical membrane pump theory (CMPT)	Brings the first electronic model for electrophysiology; more viable/tangible than a merely ion-centric view.	[1,19,78]
10	Rotary mechanoproteins & bacterial flagella-aided motility (mechanical work)	Berg’s & Boyer’s rotary model	Disclaiming the rotary model for flagellar motility, a facile water-ejection based propulsion was proposed; agreeing with Type 3 secretion system structure.	[1,4,11,13,14,15,78,86]
11	Unusual dose responses (idiosyncrasies and hormesis), photodynamic therapy, affects in chemically disconnected system, etc.	No explanation!	Reasons out non-genetic interpersonal variabilities, enhanced activity at lower doses in some cases. Could offer novel ways to approach the unusual effects of temperature, electrostatics, etc. on life.	[1,5,14,44,77,111] & this work!
12	PCHEMS to PTMs (post-translational modfications; cancer, biological intelligence and OSTEoL)	No equivalent!	In conjunction with the central dogma, brings key aspects of molecular and macroscopic cellular activities under a unified perspective.	[1,5,11,14,15,19,20,75,77,78,111]

* More specific references in each field available at KMM’s Google Scholar profile (unlisted to avoid overt “self-citations”).

**Table 5 ijms-26-07542-t005:** A compilation of various radiations (internally generated or externally incident) and their interactive outcomes with molecular systems in cells.

Radiation Type *	Wavelength *	Frequency *	Energy * kJ/mol	Known/Projected Cellular Effects
*Near UV (UV-A)*	315–400 nm	750–950 THz	302–382	Activates flavins, induces DROS productions, affects mitochondria
*Visible Light*	400–700 nm	430–750 THz	170–300	Photoreception (opsins), chlorophyll activation, circadian rhythms, ROS modulation, optogenetics, photodynamic therapy, mitochondrial stimulation
*Near Infrared (NIR)*	700–1400 nm	214–430 THz	85–170	Mitochondrial cytochrome c oxidase activation, alters mitochondrial TMP, photobiomodulation, wound healing, ATP production
*Mid-Infrared (MIR)*	1400–3000 nm	100–214 THz	40–85	Vibration excitation of bonds, localized heating, influence on protein folding and conformational changes
*Far Infrared (FIR)*	3–1000 µm	0.3–100 THz	1.2–40	Low-level thermal effects, water absorption influencing its structuring and hydrogen bonds, modulation of membrane fluidity
*Microwaves*	1 mm–30 cm	1–300 GHz	0.004–1.2	Membrane potential alteration, Ca^2+^ channel effects, dielectric heating, signal modulation, membrane polarization
*Radiofrequency (RF)*	30 cm–10 km	30 kHz–1 GHz	<0.004	Potential long-term effects on signaling, orientations, etc.
*Extremely Low Frequency (ELF)*	>10 km	3–30 kHz	<<0.001	Effects on ion channels, circadian rhythms, potential stress response triggers, EM field and electroreception

* The wavelength/frequency/energy ranges [112,113,114] and physiological/chemical effects are compiled based on various sources on the internet.

## Data Availability

All data required to peruse this work are presented within the manuscript itself.

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
