# Peer review of "Murburn Bioenergetics and “Origins–Sustenance–Termination–Evolution of Life”: Emergence of Intelligence from a Network of Molecules, Unbound Ions, Radicals and Radiations"

_ijms, 2025, doi:10.3390/ijms26157542_

Round 1
Reviewer 1 Report
Comments and Suggestions for Authors
This manuscript is about “murburn concept” and bioenergetics. I have several questions and suggestions for the authors.
1. Maybe the template is not the right version. The line numbers are missing for this manuscript. The author might want to use the newest version of the template for this journal.
2. On page 2, the section names “Introduction” and “Conventional views of life” have the same font size. If the “Conventional views of life” belong to the “Introduction” section. The authors might want to use different font sizes for section name and subsection name.
3. For the “Introduction” section and “Conventional views of life” section, some references are necessary. However, no references have been included in these two sections.
4. For Figure 1 (top panel), the authors claimed that the top panel demonstrates the classical perspective, however, the reference is missing.
5. In table 1, there are many data points, however, the references are missing.
6. The contributions of the two authors are not clarified.
Author Response
This manuscript is about “murburn concept” and bioenergetics. I have several questions and suggestions for the authors.
We thank the reviewer for taking the time out and giving us inputs. We are not really sure why the ratings for our manuscript are low for the various heads, as the comments are only suggestive. We have made honest attempts to improve the writing based on the inputs of this and the other referees.
- Maybe the template is not the right version. The line numbers are missing for this manuscript. The author might want to use the newest version of the template for this journal.
We submitted our manuscript in the free-format option, which was converted into the template at the editorial end. The revision is also submitted in the free-format option, as we thought typesetting is better left to the professionals at MDPI. We have improved the language to enhance comprehension.
- On page 2, the section names “Introduction” and “Conventional views of life” have the same font size. If the “Conventional views of life” belong to the “Introduction” section. The authors might want to use different font sizes for section name and subsection name.
We accept the reviewer’s input and we have duly revised the formatting of the section/subsections.
- For the “Introduction” section and “Conventional views of life” section, some references are necessary. However, no references have been included in these two sections.
We have now introduced some references in these two sections, as per the reviewers’ inputs.
- For Figure 1 (top panel), the authors claimed that the top panel demonstrates the classical perspective, however, the reference is missing.
We have converted Figure 1 into two figures (separated the panels!). and introduced references.
- In table 1, there are many data points, however, the references are missing.
The table comprises of impressions gathered from dispersed literature. The energy bandwidths are common knowledge. We have included some references as per the reviewer’s inputs.
- The contributions of the two authors are not clarified.
We have detailed authors’ contributions and hope that the changes (duly highlighted) are acceptable.
Reviewer 2 Report
Comments and Suggestions for Authors
- While the paper mentions some experiments, most of it is based on theory. It would be stronger if it included clear predictions and more experimental evidence to support the ideas.
- The paper strongly criticizes existing models like chemiosmosis and the electron transport chain, but it doesn’t provide enough solid evidence to fully back up this criticism.
- Some parts are hard to follow and seem speculative. It would help to clearly separate what is well-proven from what is still a hypothesis.
- The paper should be better organized, with clearly marked sections that explain the theory, show the supporting evidence, and discuss the implications.
- The authors should avoid making very strong or absolute statements unless they have strong evidence. Words like “untenable” or “only model” should be used carefully.
- Much of the paper relies on the authors’ previous work. It would be helpful to show support from other independent studies to confirm their claims.
Author Response
We thank the reviewer for appreciating the core aspects of our work (65% rating) and language, and giving us inputs to enhance the presentation/perceptions thereof. We hope that our efforts to address the points raised and revision made are appreciable.
- While the paper mentions some experiments, most of it is based on theory. It would be stronger if it included clear predictions and more experimental evidence to support the ideas.
Manoj et al. (2018) Archives of Biochemistry and Biophysics makes a 30-point predictability comparison between the “ETC-chemiosmosis-Rotary ATP synthesis” model and the murburn model. However, based on the reviewer’s input, and given the time lapsed and developments thereof, we have included three elaborate tables in Section 4 to make point-wise comparison between various aspects (experimental, theoretical, structure-distribution, evolution, etc. of the two bioenergetic models. We hope the update is appreciable. The reviewer can also note the Table in Section 9, which shows the multiple/diverse areas where murburn concept has been applied; and these are published in reputed journals.
- The paper strongly criticizes existing models like chemiosmosis and the electron transport chain, but it doesn’t provide enough solid evidence to fully back up this criticism.
To the best of our beliefs, we have conclusively disclaimed the acclaimed model and proposed a viable alternate, which agrees with all available information and observation reported till date on the system (our earlier publications in the field are testament to this statement). With the detailed revision (inclusion of an elaborate point-wise comparison tables), we hope our legitimate abrogative interest/pursuit would be evident. (The problem is that if we cite all our papers in this regard, it will be called self-citation and Editors/Reviewers raise flags!)
- Some parts are hard to follow and seem speculative. It would help to clearly separate what is well-proven from what is still a hypothesis.
We have made attempts to clarify issues in the revision and hope things are better now. If the reviewer is more specific, we could further make improvements.
- The paper should be better organized, with clearly marked sections that explain the theory, show the supporting evidence, and discuss the implications.
We have marked sections and discussed more elaborately to project the implications of the new ideas.
- The authors should avoid making very strong or absolute statements unless they have strong evidence. Words like “untenable” or “only model” should be used carefully.
We understand the seriousness and gravity of our claims. We have systematically pursued an agenda for decades and we now have conclusive evidence/arguments to use words like “untenable”. We are very particular of the negation-intent in our pursuit. However, we are quite comfortable with removing exclusive claim for the new model (even if it is true!).
- Much of the paper relies on the authors’ previous work. It would be helpful to show support from other independent studies to confirm their claims.
It is unfortunate that the ideas that we disclaim are acclaimed and enshrined in the textbooks that we learned and/or taught from. The philosophy and practise of science dictate that we should make amends when the falsity is evident. In this regard, the main point/claim of our pursuit is to point out that earlier theoretical formulations and interpretations were misplaced. We don’t see any data or observation that does not agree with the new model proposed, as you can see from the three Tables. We are at a loss to bring about support from people outside our group; we can only take people to reason but can’t enforce them to re-track.
Once again, we thank this reviewer for the inputs aimed at improving our manuscript and hope that the changes made (duly highlighted) are appreciable/acceptable.
Reviewer 3 Report
Comments and Suggestions for Authors
The submitted review on topic is one of the other several review on murburn concept by one of author (Kelath Murali Manoj). Overall the review is good. However, I have certain suggestion that would make it better for readers of the review. My specific comments/suggestions are:
1: Abstract needs rewiring since it is very confounding in current form and does out bring out the message about content on the review or fundamental of the concept discussed in review. It is needed to be concise and lay out the message of topic more clearly.
2: Since murburn concept is not very widely studied in the biological field (to my knowledge there are very few research papers on topic other than those by one of the authors), the basic fundamental concept of murburn needs to be introduced in more better and simpler form so as to explain the concept for general readers. Readers should get more better idea of the murburn concept and more specifically in biology.
3: Authors should include a paragraph on the role of murburn concept in light of epigenetics and cancer and write how does murburn concept help in explaining some of epigenetic outcomes.
4: Authors should include a paragraph commenting on acceptability of murburn concept in biological field and comment why this concept has not been explored in research more widely and why it is necessary for biochemical scientists or cancer biologist to adapt and use murburn concept. The new paragraph in review can also have content on criticism and limitations of murburn concept.
Author Response
The submitted review on topic is one of the other several review on murburn concept by one of author (Kelath Murali Manoj). Overall the review is good. However, I have certain suggestion that would make it better for readers of the review. My specific comments/suggestions are:
We thank the reviewer for appreciating the core aspects of our work, seeing positives in the write-up and for giving us inputs to enhance the presentation/perceptions of the manuscript. Wherever possible, we have made to improve/clarity (whether in continuity or language). We hope our efforts are appreciable. We proceed to address the issues raised point-wise.
1: Abstract needs rewiring since it is very confounding in current form and does out bring out the message about content on the review or fundamental of the concept discussed in review. It is needed to be concise and lay out the message of topic more clearly.
We have re-written the abstract to highlight the fundamental of the concept discussed in the review. We hope that the new writing (with a more precise title) captures the essence more clearly and succinctly.
2: Since murburn concept is not very widely studied in the biological field (to my knowledge there are very few research papers on topic other than those by one of the authors), the basic fundamental concept of murburn needs to be introduced in more better and simpler form so as to explain the concept for general readers. Readers should get more better idea of the murburn concept and more specifically in biology.
We appreciate this input and we have taken some space to detail the basic fabric of murburn concept, how it differs from erstwhile ideas, and what clarity and impact it is poised to bring in biology and medicine. The introduction is longer, and there are 4 new tables introduced. We hope the revised version appeals to the reviewer.
3: Authors should include a paragraph on the role of murburn concept in light of epigenetics and cancer and write how does murburn concept help in explaining some of epigenetic outcomes.
We have duly added the required paragraph (Section 9E) on the role of murbun in epigenetics and cancer (the latter was already discussed). We thank the reviewer for this suggestion.
4: Authors should include a paragraph commenting on acceptability of murburn concept in biological field and comment why this concept has not been explored in research more widely and why it is necessary for biochemical scientists or cancer biologist to adapt and use murburn concept. The new paragraph in review can also have content on criticism and limitations of murburn concept.
We have done what the reviewer seeks of us (Section 10), as it is an important aspect of the stature we experience now (as few seem to get ready to check out our ideas!). Actually, one of us had asked the question why to ChatGPT (and the outcome it gave is published in the latest version of Biomedical Reviews, the 3.5 decades old official annual publication of the Bulgarian Society of Cell Biology). We have included a brief paragraph on the above aspect and addressed the specific question why murburn concept should be adopted by biochemists and biologists and medical practitioners.
We do not have anything to criticize regarding murburn; other than that it does not appeal to aesthetics of researchers inclined to see DRS as mere agents of chaos. Neither do we see/present any limitation of murburn. (Reality is what it is!) This is because it is a concept (a summation of facts and direct consequences thereof), not a hypothesis. (This is reflected in the new abstract.) We can of course, comment on the limitation of murburn models for various systems but we believe that would be beyond the context of the current write-up.
We thank this open-minded and constructive reviewer for the inputs; and hope that the changes made in the revision are acceptable.
Round 2
Reviewer 3 Report
Comments and Suggestions for Authors
The authors have revised the manuscript significantly and I am satisfied with the revision.